# A Review of the Genus *Cloeon* from Chinese Mainland (Ephemeroptera: Baetidae)

**DOI:** 10.3390/insects12121093

**Published:** 2021-12-07

**Authors:** Xiaoli Ying, Wenjuan Li, Changfa Zhou

**Affiliations:** The Key Laboratory of Jiangsu Biodiversity and Biotechnology, College of Life Sciences, Nanjing Normal University, Nanjing 210023, China; yingxiaoli9106@163.com (X.Y.); 18356518906@163.com (W.L.)

**Keywords:** mayfly, lentic environment, taxonomy, emerge time, key

## Abstract

**Simple Summary:**

The mayflies of baetid genus *Cloeon* Leach, 1815 are common inhabitants of lentic environments, such as ponds, lakes, reservoirs, and even some temporary bodies of water near human communities. Therefore, they are frequently collected and comprehensively researched around the world. In China, however, although nine *Cloeon* species names have been reported, few of them are confirmed historically. In this study of over 1000 specimens from over 50 sites on the Chinese mainland, four species of *Cloeon* were identified, namely *C. bicolor* Kimmins, 1947, *C. dipterum* (Linnaeus, 1761), *C. harveyi* (Kimmins, 1947), and *C. viridulum* Navás, 1931. The record of *C. bicolor* is the first in China, and its nymphs are described for the first time. Three junior synonyms are proposed for the species *C. viridulum*. The present research is not only a revised review of the genus *Cloeon* for the Chinese mainland but also a solid base for future works on Chinese and Asian *Cloeon* taxonomy.

**Abstract:**

The widely distributed and species diverse genus *Cloeon* Leach, 1815 has never been reviewed in China, although nine species names have been reported from this country. After checking types of two species (*C. viridulum* and *C. apicatum*) and newly collected materials from more than 50 sites, four species are recognized, compared, and photographed in this research. Among them, the species *C. bicolor* Kimmins, 1947 is first recorded in China, and its nymphs are described for the first time. The distribution in China of two other species (*C. dipterum* (Linnaeus, 1761) and *C. harveyi* (Kimmins, 1947)) are also confirmed. The two species established by Navás (*C.*
*apicatum* Navás, 1933 *= C. navasi* Bruggen, 1957, *C. pielinum* Navás, 1933, syn. nov.) and the species *C. micki* named by Tong and Dudgeon in 2021 are synonymized with *C. viridulum* Navás, 1931—the fourth species in this study. Furthermore, the *C. virens* Klapálek, 1905 from the Chinese mainland, found by Ulmer in 1925, is also regarded as the last species. Among the four species, the *C. dipterum* and *C. harveyi* seem closer because of the similar female color patterns of their stigma and ventral abdomen, while the species *C. viridulum* and *C. bicolor* are more alike because they have neither pigmented stigma nor stripes on abdominal sterna.

## 1. Introduction

*Cloeon* Leach, 1815 [1] (or Cloeon/fg2 *sensu* Kluge, 2012 [2]) is a cosmopolitan mayfly genus [3,4]. Diverse species have been reported from Africa [5], Europe [6], North America [7], South America [8,9], India [10,11], Japan [12], Korea [13], Mongolia [14], Australia [15], and Pacific Islands [16]. However, the species in this genus have never been systematically studied in China.

Historically, nine *Cloeon* species names have been reported from China [4,17,18]. Among them, eight were described by Navás and Ulmer between 1910–1940. They describe subimagoes, females, or even undeclared sexes, but the types are lost or damaged [19,20] and the descriptions are usually extremely brief and often lack figures. Consequently, these poorly described species remain an impediment to a clear understanding of the species found in China. The ability to recognize new species (e.g., *Cloeon micki* Tong and David, 2021 [4]) and those previously described is not clear.

In order to solve this problem, we collected and reared *Cloeon* species from more than 50 water bodies on the Chinese mainland. We examined photos of type material (*C. viridulum* and *C. apicatum*) from Museo de Zoologia del Ayuntamiento, Barcelona, Spain and some specimens in Heude Museum deposited by Navás and Hsu in 1930. From these materials, we were able to recognize four species from the Chinese mainland, which were photographed and descriptions were updated. Unfortunately, three species from the Taiwan province, China (*C. virens* Klapálek, 1905, *C. bimaculatum* Eaton, 1885, *C. marginale*, Hagen, 1858) [18,21] were limited in our collections, and further material is required to determine their real status in China.

## 2. Materials and Methods

The larvae were collected by hand net sweeping and filtering aquatic plants in water. Mature larvae were reared in a plastic container with natural water covering a nylon net indoors. Some adults were attracted by LED lights near a creek and pond. All materials were stored in ethanol (around 85%).

All specimens were photographed with a digital camera (Single Lens Reflex, Guangzhou, China) and examined under a stereo microscope (Mingmei Photoelectric, MZ81, Guangzhou, China). Some small structures, such as mouthparts, claws, and gills, were placed on temporary slides with ethanol to observe and photograph with a microscope camera (Nikon eclipse 50i, Japan).

All specimens used in this study are deposited in the Mayfly collection, College of Life Sciences, Nanjing Normal University (NJNU), China.

## 3. Results

### 3.1. Cloeon bicolor *Kimmins, 1947 (First Record from China)*

*Cloeon bicolor* Kimmins, 1947: 97, figures 4, 8, 12 (male, female) [10]. Type: Female, Calcutta, Bengal.

*Cloeon bicolor*: Gillies, 1949: 173 [22]; Kimmins, 1971: 311 [23]; Hubbard and Peters, 1978: 8 [11].

Distribution: China (south); Bengal, India.

Descriptions: Adults (see Kimmins, 1947 [10]).

Mature nymphs (first description): body length 5.0–7.0 mm, caudal filaments 3.0–4.0 mm, terminal filament slightly shorter than cerci (Figure 1A,B). Body generally brown with various dark dots and markings dorsally (living nymphs usually greenish brown) (Figure 1A), ventral part of body pale (Figure 1B). Midline of dorsal body pale, one pair of brown stripes along with midline on vertex (Figure 1A). Antennae 2.0–3.0 mm, scape and pedicle brown, flagella pale; scape and pedicle with tiny sparse hair-like setae on surface, flagella with tiny setae on articulations (Figure 2D). Compound eyes and ocelli dark brown. Mouthparts: dorsal surface and free margins of labrum with hair-like setae, ventral surface with two tufts of hair-like setae near median emargination; leading margin of labrum slightly convex, median notch emarginated smoothly (Figure 3A); mandibles as in figures (Figure 2G,H), lateral margin with very sparse hair-like setae, prostheca strong, with a finger-like denticle and a tuft of spines. Maxillary palpi with very sparse hair-like setae on surface, second and third segments subequal in length, both of them shorter than first segment (Figure 3E). First segment of labial palpi slightly longer than second, the latter longer than third, which is slightly expanded (Figure 3I); all segments with sparse hair-like setae on surface, those of apical segment longer and denser than others; glossae and paraglossae with relatively long setae on free margins. Lingua of hypopharynx round, free margins of lingua and superlinguae with hair-like setae (Figure 2I).

Nota with irregular brown markings. Apical half of trochanter brown, femora of all legs pale but with subapical brown band and brown base, tibiae and tarsi with bands near bases (Figure 2A–C). Legs with hair-like setae on surface, inner and outer margins; besides those setae, femora of all legs with pectinate setae on inner margin, spatulate setae on dorsal surface and spine-like setae on outer margin; setae patterns of tibiae and tarsi similar to femora except without any spine-like setae on outer margins. Patellar–tibial fusion lines present on all legs; all legs subequal in length; length of femora: tibiae: tarsi of foreleg = 1.5:0.82:1.0, length of ratio of midleg = 1.8:1.0:1.0, length of ratio of hindleg = 1.83:1.0:1.0. All claws similar, with slightly expanded base and two rows of spine-like denticles at basal half, denticles progressively larger from base to apex (Figure 3M).

Besides pale midline of abdominal terga, each tergum with three pairs of pale dots (Figure 1A): one pair near midline, one on anterolateral angles and one on posterolateral angles. Median pair round pale dots relatively regular but other two pairs irregular in size and shapes. Terga IV and VII paler than others. Lateral margins of posterior half tergum VIII and whole tergum IX with spines (Figure 2F); posterior margins of terga with spines. Paraproct with spines on inner margin except 1/3 base (Figure 2J). Gills I–VII with clear dark tracheae (Figure 4A–G), gills I and II with reddish pigments at base (color very clear in living nymphs) (Figure 4A,B); gills I–VI with double lamellae but gills VII single (Figure 4A–G); ventral one of gills I–VI nearly round to heart-shaped, but dorsal lamellae of gill I elongated to leaf-like in shape (Figure 4A), dorsal lamellae of gills II heart-shaped with expanded basal lobe (Figure 4B), those of gills III–VI near round (Figure 4D–F); ventral lamellae of gills II–VI with slightly sclerotized outer half margins; both inner and outer margins of gills VII slightly sclerotized (Figure 4G), hard line of outer margin broader and longer than that of inner one (Figure 4I,J); all free margins of all gills with very sparse tiny hair-like setae, but those of gills VII slightly denser and more numerous (Figure 4H). Caudal filaments pale, every four segments with a brown ring (Figure 1A,B and Figure 2E); basal 2/3 cerci with long hair-like setae on mesal margins but terminal filament with similar setae on both lateral margins; all articulations of caudal filaments with a ring of spines, those on outer margins of cerci longer and broader (Figure 2E).

Diagnosis (based on specimens stored in alcohol): The male imagoes of this species can be identified by their transparent wings (Figure 5A and Figure 6A), almost chestnut to reddish terga (Figure 5A and Figure 7A), each tergum with a pale midline and a pair of triangular to round pale dots (Figure 7A); pale portions are usually enlarged and emerged into a larger portion in each tergum, producing a wide longitudinal pale midline; lateral margins of terga also pigmented with short conspicuous reddish stripes (Figure 7A,B); abdominal sterna washed with irregular light reddish pigments or stains at midlines, those of sterna VII–IX sometimes forming reddish dots (Figure 7B); a round projection with a round tip between two forceps; segment 2 of forceps with slightly expanded base and apex; segment 3 tiny (Figure 7C). The female imagoes can be differentiated by their almost reddish terga (Figure 5B and Figure 8A), each tergum with a pale midline and a pair of pale dots (Figure 8A); lateral margins of terga also with reddish lines (Figure 8A,B); abdominal sterna pale but midline of sterna VII–IX usually with dots on median posterior margins (Figure 8B). Wings of females have very conspicuously pigmented costal and subcostal sections (Figure 5B and Figure 6B).

Remarks: Mature nymphs of this species are similar to *C. harveyi* (Figure 1E,F) and *C. viridulum* (Figure 1G,H) by having terga VIII and IX with lateral spines and caudal filaments without median dark bands (Figure 1A,B) (in contrast, *C. dipterum* have lateral spines on terga VI–IX and dark markings on caudal filaments, Figure 1C,D). However, the nymphal stages of *C. bicolor* can be identified by pale abdominal sterna with median reddish dots on posterior margin only (those of *C. viridulum* are totally pale, abdominal sterna of *C. harveyi* have clear longitudinal stripes) (Figure 8A,E,G), the length of claws (longer than *C. viridulum* but shorter than *C. harveyi*) (Figure 3M,O,P), and relatively regular pairs of pale dots on terga. The nymphs of this species are very similar to *C. viridulum* in color pattern and most structures (such as the claw, maxilla, lateral spines on terga VIII–IX), but they have relatively dark bodies (especially those terga II–III, V–VI) but lack independent dark dots on terga II and V (Figure 1A,G).

The six indoor-reared nymphs were seen molting into subimagoes at 18:00–20:00 p.m. local time. Before 7:00 a.m. the next morning, they became imagoes.

Materials examined: Two male imagoes and eight female imagoes, Benhao Town, Lingshui Li Autonomous County, Hainan Province (109.962463° E, 18.608751° N), 17-III-2021, collected by Mei Xiong; six female imagoes, Nanxing Farmhouse, Baisha Li Autonomous County, Hainan Province (109.4429° E, 19.221641° N), 12-III-2021, collected by Mei Xiong; one male imago and three female imagoes, Maozhan Village, Maoyang Town, Wuzhishan City, Hainan Province (109.515944° E, 18.944574° N), 24-VII-2014, collected by Jianhua Dai, Dan Zhou, Junzhi Sun; two male imagoes and three female imagoes, Diaoluoshan Forestry Bureau, Lingshui Autonomous County, Hainan Province (109.933185° E, 18.662679° N), 22-VII-2014, collected by Jianhua Dai, Dan Zhou, Junzhi Sun; one male imago and two female imagoes, Wanquan River, Longjiang Overseas Chinese High School, Longjiang Town, Qionghai City, Hainan Province (110.326796° E, 19.147609° N), 16-VII-2014, collected by Jianhua Dai, Dan Zhou, Junzhi Sun; one female imago, Sixin Sawmill, Wuyi Mountain, Fujian Province (117.741085° E, 27.600284° N), 6-VII-2021, collected by Dewen Gong; one female imago, Yanzhang Town, Longquan City, Lishui City, Zhejiang Province (119.012373° E, 28.186258° N), 9-VII-2021, collected by Dewen Gong; two male imagoes and four female imagoes with 30 nymphs, Xiamen Botanical Garden, Fujian Province (118.109629° E, 24.4479° N), 18-20-X-2021, collected by Xiaoli Ying, Wenjuan Li.

### 3.2. Cloeon dipterum *(Linnaeus, 1761)*

*Ephemera diptera* Linnaeus, 1761: 377 [24]. Types: adult, Europe.

*Caenis sinensis* Walker, 1853: 584 [25] (male) (first record from China, synonymized by Kimmins, 1960: 292. [26])

*Cloeon sinense*: Eaton, 1885: 189 [27]; Ulmer, 1925: 101 [28]; Ulmer, 1935–1936: 214 [21]; Wu, 1935: 250 [29]; Gui, 1985: 81 [30]. Synonymized by Kimmins, 1960: 292 [26].

*Cloeon dipterum*: Leach, 1815: 137 [1]; Ulmer, 1929: 23, figures 82, 83, 85, 86, 146 [31]; Ulmer, 1935–1936: 211, 214 [21]; Imanishi, 1940: 215, figure 29 [32]; Sowa, 1975: 215, figures 1, 5, 7, 10–12 [33]; You and Gui, 1995: 33, figure 28 (male) [34]; Bae and Park, 1997: 304, figures 1–4, 17–19 (nymph, female, male) [13]; Bae, 1997: 406 [35]; Quan et al., 2002: 244, figures 7, 58, 183 (nymph) [36]; Gattolliat et al., 2008: 52 [3].

For more citations and complete synonymy, see Bauernfeind and Soldán, 2012 [6].

Distribution: China (central to northern); Cosmopolitan.

For descriptions, see Bae and Park, 1997 [13], Bauernfeind and Soldán, 2012 [6] or Sowa, 1975 [33].

Diagnosis (based on specimens stored in alcohol): The male imagoes of this species can be identified by their pairs of crescent to triangular reddish brown markings on abdominal terga (Figure 7D); posterior half of terga I–III and midline of terga VI–IX are alsoreddish (Figure 7D); lateral margins of terga are reddish (Figure 7D); pairs of reddish near-rectangular markings present on abdominal sterna (Figure 7E). Wings lacking any pigments except at the base of subcostal brace (Figure 6C). A cone-like median projection presents between forceps or on penial bridge (Figure 7F). The female imagoes can be differentiated by their similar marking patterns to males (Figure 8C and Figure 7D), the heavily pigmented C and Sc sections of wings with irregular pale blanks (Figure 6D). The mature nymphs of this species can be recognized by their relatively short apical segments of maxillary palpi (Figure 3F), spines at least present on lateral lines of terga VII–IX, base of claw slightly broadened and basal 1/2 of claw with two rows of denticles (Figure 3N).

Remarks: Both Gattolliat et al. (2008) [3] and Bauernfeind and Soldán (2012) [6] stated clearly that the nymphs of European *C. dipterum* or complexes of this species have spines on terga V–IX and the basal 2/3 of tarsal claws. However, all our collections from China and several nymphs from Vladivostok (Russia) have spines on the lateral margins of terga VII–IX only, although some individuals have three spines on terga VI. In addition, the tarsal claws have denticles on the basal half (Figure 3N). Nevertheless, considering that Russia, Korea, and Japan have this species [12,13,37], we treat these two differences as variations between populations.

Materials examined: One male imago with five nymphs, Daqinggou, Tongliao, Inner Mongolia Autonomous Region (111.588869° E, 40.811141° N), 29-VIII-2014, collected by Li Shi, Mingrun Tian, Yuxuan Zhu, Xuefeng Gao, Weijie Sun; 20 nymphs, Municipal Committee of Jianshe Road, Qingshan District, Baotou City, Inner Mongolia Autonomous Region (109.953208° E, 40.621117° N), 13-VIII-2014, collected by Qi Si; three male imagoes and five female imagoes with 50 nymphs, Shengziping Village, Renheping Town, Wufeng County, Hubei Province (111.246234° E, 30.10159° N), 11-14-VII-2013, collected by Dan Zhou; 33 nymphs, Hexi bridge, Qianjiadian Town, Yanqing District, Beijing (116.346835° E, 40.693859° N), 1-6-VIII-2017, collected by Wei Zhang, Zhenxing Ma; 15 nymphs, small pond in the Far Eastern Federal University of Vladivostok, Russia, 24-VIII-2016, collected by Changfa Zhou, Xiaoyan Shao.

### 3.3. Cloeon harveyi *(Kimmins, 1947)*

*Procloeon harveyi* Kimmins, 1947: 94, figures 2, 6, 10 (male, female) [10]. Type: female, Calcutta, Bengal.

*Procloeon harveyi*: Gillies, 1949: 176 (first record from China, Hong Kong) [22]; Kimmins, 1971: 314 [23]; Hubbard and Peters, 1978: 11 [11]; Hubbard and Srivastava, 1984: 3 [38]; Kandibane et al., 2007: 743 [39]; Mukherjee et al., 2012: 55 (figures 1–9, all stages) [40].

*Cloeon harveyi*: Müller-Liebenau and Hubbard, 1986: 538 [41]; Hubbard, 1986: 248 [42].

Distribution: China (south); Malaysia, India, Thailand.

For descriptions, see Kimmins, 1947 [10] and Mukherjee et al., 2012 [40].

Diagnosis (based on specimens preserved in alcohol): The male imagoes of this species can be identified by their terga I, VII–IX reddish brown, terga II–III, V–VI with clear triangular dots (those of tergum V sometimes very small) (Figure 7G); posterior and lateral margins of terga also reddish brown (Figure 7G); at least abdominal sterna VII–IX with a pair of longitudinal reddish stripes (Figure 7G), those of VIII–IX maybe fused together (Figure 7H); wings totally transparent except the subcostal brace (Figure 6E); cone-shaped median projection on penial bridge (Figure 7I). The female imagoes can be differentiated by their similar abdominal marking patterns to males, but the markings of abdominal sterna are more conspicuous (Figure 8E,F); wings have heavily pigmented subcostal brace and pterostigma (Figure 6F). The mature nymphs of this species can be diagnosed by their relatively slim maxillary palpi (Figure 3G) and slender claw (Figure 3O), three segments of maxillary palpi subequal in length (Figure 3G), slightly broadened apical segment of labial palpi (Figure 3K), spines present on lateral lines of terga VIII–IX only, base half of claw with denticles (Figure 3O). The color pattern of mature nymphs is also similar to adults (Figure 1E,F and Figure 8E,F).

Remarks: The nymphs of this species can be collected from or found sharing the same pool or pond together with *C. viridulum* and *C. bicolor.* Three of them were found collecting from same water at several places.

Materials examined: 3 male imagoes and 10 female imagoes with 20 nymphs, Dinghai Park, Dinghai County, Zhoushan City, Zhejiang Province (122.106916° E, 30.0124° N), 14-15-IX-2014, collected by Dan Zhou, Yike Han, Junzhi Sun; 4 nymphs, County north reservoir upstream stream, Dinghai County, Zhoushan City, Zhejiang Province (122.09827° E, 30.013398° N), 15-IX-2014, collected by Dan Zhou, Yike Han, Junzhi Sun; 20 nymphs, Downstream creek Tropical Rainforest Scenic Area, Shuiman Town, Wuzhishan City, Hainan Province (109.678087° E, 18.842413° N), 7-I-2015, collected by Changfa Zhou, Yike Han, Junzhi Sun; 15 nymphs, Fish pond in Shanei Village, Ding’an County, Hainan Province (110.265036° E, 19.687255° N), 5-I-2015, collected by Changfa Zhou, Yike Han; 1 nymph, Dasong Mountain Forest Park, Yichang City, Hubei Province (111.351855° E, 30.439953° N), 16-VII-2013, collected by Yanxia Wang, Dan Zhou; 1 nymph, Jiudaohe Village, Zhicheng Town, Yidu City, Hubei Province (111.493714° E, 30.215748° N), 17-VII-2013, collected by Yanxia Wang, Dan Zhou, Chuanren Li.

### 3.4. Cloeon viridulum *Navás, 1931*

*Cloeon viridulum* Navás, 1931: 7, figure 14 (female subimago) [43]. Type: female subimago, Jiangsu = Kiangsu, Shanghai.

*Cloeon viridulum*: Navás, 1933a: 17 [44]; Wu, 1935: 251 [29]; Ulmer, 1935–1936: 215 [21]; Sun et al., 2015: 58 (habitat) [45]; Si et al., 2017:1 (transcriptome) [46].

*Cloeon apicatum* Navás, 1933a [44]: 17 (*nec* 1932: 21 [47]). Types: female and male imagoes, Chusan, Zhejiang province (renamed as *Cloeon navasi* by van Bruggen, 1957: 37 [48]). **Syn. Nov.**

*Cloeon apicatum*: Wu, 1935: 250 [29]; Ulmer, 1935–1936: 214 [21]; Gui, 1985: 82 [30].

*Cloeon navasi*: Alba-Tercedor and Peters, 1985: 221 [20].

*Cloeon micki* Tong and Dudgeon, 2021 [4]: 1, figures 1–31. Types; adults and nymph, from Hong Kong and Guangdong, China. **Syn. Nov.**

*Cloeon pielinum* Navás, 1933b: 8, figure 42 (adult unknown sex) [49]. Type: adults, from Shanghai, China. **Syn. Nov.**

*Cloeon pielinum*: Wu, 1935: 250 [29]; Ulmer, 1935–1936: 214 [21]; Gui, 1985: 82 [30].

*Cloeon virens*: Ulmer, 1925: 100 (Guangdong Province, China) [28]; Ulmer, 1935–1936: 211, 214 (Peiping = Beijing, Soochow = Suzhou, Kiangsi = Jiangxi, Kiangsu = Jiangsu) (*nec* Klapálek, 1905: 106, misidentification) [21]; Wu, 1935: 251 [29]; Gul, 1985: 82 [30]; You and Gui, 1995: 36, figure 31 (male) [34]; Gui et al. 1999: 327, figure 11-5 [50].

*Cloeon bicolour* [*sic*.]: Kubendran et al., 2017: 617 [51] (*nec Cloeon bicolor* Kimmins, 1947 [10], mis-identification).

Distribution: China (most regions except northeast and northwest); India.

For description, see Tong and Dudgeon, 2021 [4].

Diagnosis (based on specimens stored in alcohol): The male imagoes of this species can be identified by their pale terga I–VII and abdominal sterna but terga VIII–IX reddish brown (Figure 7J,K), wings transparent (Figure 6G); round projection between forceps (Figure 7L). The female imagoes of it can be differentiated by its orange to reddish dorsal terga, each tergum with a pair of short bent pale stripes dorsally, lateral margins of terga pigmented with reddish dots (Figure 8G); pale abdominal sterna (Figure 8H). Wings transparent but costal and subcostal sections yellowish and semi-hyaline (Figure 6H). The mature nymphs of this species can be diagnosed by their three subequal segments of maxillary palpi (Figure 3H), without an expanded apical segment of labial palpi (Figure 3L), relatively short tarsal claw with denticles on basal 2/3 (Figure 3P). Most nymphs have two conspicuous dark spots on terga II and V, respectively (Figure 1G).

Remarks: This species can be found in most parts of the Chinese mainland, including some man-made habitats, such as ponds in school campuses, city gardens, or reservoirs.

Based on pictures of specimens studied by Navás in 1931 and 1933, the species *C. apicutum* is regarded as a junior synonym of *C. viridulum*. The types of both species were collected from the same region (Shanghai, Jiangsu to Zhejiang province, eastern China), in which we intensively collected specimens.

Although the types of *C. micki* Tong and Dudgeon have not been examined, the presence of two spots on abdominal terga II and V in all life stages (occasionally not obvious), the similar male genitalia, dorsal color pattern of males and females, and characteristics of the tarsal claws and mouthparts of the nymphs support the designation of *C. micki* as a junior synonym of *C. viridulum*.

Due to the wide distribution and frequent collection of *C. viridulum*, the species *C. virens* found by Ulmer (1925, 1935–1936) on the Chinese mainland is also recognized as this species here. We found no individuals similar to *C. virens* in this region.

The type of the species *C. pielinum* is lost and without sex declaration. Based on its equal distribution (type from Shanghai) and similar description to the *C. viridulum*, it is also synymonized here.

Materials examined: 20 male imagoes and 18 female imagoes with 68 nymphs, Caiyue Lake, Xianlin Hotel, Nanjing Normal University, Jiangsu Province (118.91025° E, 32.106838° N), 13-24-IV-2014, collected by Dan Zhou, Yike Han, Junzhi Sun; 18 male imagoes and 12 male subimagoes with 35 nymphs, Caiyue Lake, Xianlin Hotel, Nanjing Normal University, Jiangsu Province (118.91025° E, 32.106838° N), 13-18-IX-2015, collected by Juanyan Luo; 35 male imagoes and 22 female imagoes with 50 nymphs, Caiyue Lake, Xianlin Hotel, Nanjing Normal University, Jiangsu Province (118.91025° E, 32.106838° N), 10-20-IX-2021, collected by Xiaoli Ying; 3 male imagoes and 3 female imagoes with 2 nymphs, Sun Yat-sen Mausoleum, Nanjing City, Jiangsu Province (118.854098° E, 32.054508° N), 5-VI-2013, collected by Changfa Zhou, Dan Zhou, Yanxie Wang; 2 male imagoes and 2 female imagoes with 2 nymphs, Stone Mountain Reservoir, Jurong, Jiangsu Province (119.210094° E, 32.068913° N), 3-VII-2013, collected by Changfa Zhou, Dan Zhou, Yanxie Wang; 3 male imagoes and 5 female imagoes with 55 nymphs, Baoying Lake, Yangzhou City, Jiangsu Province (119.26708° E, 33.165469° N), 20-IV-2014, collected by Dan Zhou, Junzhi Sun, Yike Han; 1 male imago and 1 female imago with 10 nymphs, Dinghai Park, Dinghai District, Zhoushan City, Zhejiang Province (122.106916° E, 30.0124° N), 14-15-IX-2014, collected by Dan Zhou, Junzhi Sun, Yike Han; 4 male imagoes and 2 female imagoes with 60 nymphs, Liwa River, Zhongbei college, East China Normal University, Putuo District, Shanghai (121.406072° E, 31.22899° N), 12-13-IX-2014, collected by Dan Zhou, Junzhi Sun, Yike Han; 18 nymphs, Near Heyuandai, Jixi County, Xuancheng City, Anhui Province (118.578519° E, 30.067533° N), 21-VII-2021, collected by Changfa Zhou, Xiaoli Ying, Wenjuan Li, Zhiming Lei; 1 male imago with 3 nymphs, Maozhan Village Creek, Maoyang Town, Wuzhishan City, Hainan Province (109.515944° E, 18.944574° N), 24-VII-2014, collected by Jianhua Dai, Dan Zhou, Junzhi Sun; 2 male imagoes and 2 female imagoes with 20 nymphs, Fish Pond, Sun Yat-sen University, Guangzhou City, Guangdong Province (113.298395° E, 23.096729° N), 19-21-XI-2014, collected by Changfa Zhou; 2 female imagoes, G3 Aquatic Plant Cultivation Pond, Jinan University Academy of Sciences, Guangzhou City, Guangdong Province (113.349402° E, 23.133382° N), 20-III-2011, collected by Hongqu Tang.

## 4. Discussion

The males of four Chinese *Cloeon* species can be divided into two groups based on their abdominal sterna with or without reddish longitudinal stripes. The species *C. harveyi* (Figure 7H) and *C. dipterum* (Figure 7E) fall into the former category, but *C. viridulum* (Figure 7K) and *C. bicolor* (Figure 7B) belong to the latter. Similarly, the females of those four species can be divided into two groups due to their similar color patterns to males. In addition, the females of *C. harveyi* (Figure 8F) and *C. dipterum* (Figure 8D) have reddish brown stigmas, pigmented C and Sc sections of wings with clear pale dots or markings, while those of *C. viridulum* (Figure 6H) and *C. bicolor* (Figure 6B) have uniformly pigmented and semi-hyaline C and Sc sections only.

Besides color patterns on abdominal sterna, the nymphs of *C. harveyi* (Figure 3G,O) and *C. dipterum* (Figure 3F,N) have longer claws and maxillary palpi than those of *C. viridulum* (Figure 3H,P) and *C. bicolor* (Figure 3E,M).

Morphologically, four *Cloeon* species of the Chinese mainland can be identified easily. However, in most cases, the color patterns of the body and wings are used, and a few solid structures can be found in their tiny bodies for differentiation. Hopefully, molecular evidence can provide more information in future work, especially on the *C. dipterum* complex or other similar species, such as *C. viridulum* and *C. ryogokuensis* Gose, 1980 [52,53].

The *Cloeon* nymphs are typical lentic mayflies, and the imaginal stages of mayflies are thought to have weak flight abilities. However, all four *Cloeon* species found in China are also reported in other Asian countries, such as India and Japan, and *C. dipterum* is a cosmopolitan species. Some Pacific Islands are also *Cloeon* habitats. The dispersal or migration routes and biogeography of these species are interesting topics for future research.

## 5. Keys to *Cloeon* Species of Chinese Mainland

### 5.1. Male Adults

1. Abdominal sterna with reddish longitudinal stripes, at least on sterna VII–IX (Figure 7E,H)-------------2

- Abdominal sterna without clear reddish longitudinal stripes (Figure 7B,K) -------------3

2. Two stripes of sterna VII–IX progressively merged together (Figure 7H) -------------***C. harveyi***

- Two stripes of sterna II–IX parallel (Figure 7E) -------------***C. dipterum***

3. Whole abdominal terga almost reddish but with regular pairs of pale dots (Figure 7A) -------------***C. bicolor***

- Terga I–VI pale, terga VII–IX pigmented; at most median half terga with orange to reddish pigments (Figure 7J)-------------***C. viridulum***

### 5.2. Female Adults

1. C and Sc sections of wings with pigmented markings (Figure 6D,F)-------------2

- C and Sc sections of wings slightly pigmented but without clear markings (Figure 6B,H)-------------3

2. Terga II–III, V–VI with distinct pairs of reddish dots or markings (Figure 8E); two stripes of abdominal sterna VII–IX progressively merged together (Figure 8F); pterostigma area of wings pigmented (Figure 6F) -------------***C. harveyi***

- Terga II–IX with distinct pairs of crescent reddish markings (Figure 8C); two stripes of abdominal sterna VII–IX parallel (Figure 8D); all sections C and Sc of wings almost pigmented but with transparent pale dots or lines (Figure 6D)-------------***C. dipterum***

3. Whole abdominal terga almost reddish but with regular pairs of pale dots (Figure 8A)-------------***C. bicolor***

- Median anterior half terga with orange to reddish pigments (Figure 8G)-------------***C. viridulum***

### 5.3. Mature Nymph

1. Lateral margins of abdominal terga VI–IX or VII–IX with spines-------------***C. dipterum***

- Lateral margins of abdominal terga VIII–IX with spines (Figure 2F)-------------2

2. Claw very slim, without clear expanded base (Figure 3O)-------------***C. harveyi***

- Base of claw expanded (Figure 3M,P)-------------3

3. Terga II–III, V–VI darker than others; terga II and V without independent median dark dots (Figure 1A)-------------***C. bicolor***

- Terga I–IX almost uniform in color; terga II and V usually with independent median dark spots (Figure 1G)-------------***C. viridulum***

## Figures and Tables

**Figure 1 insects-12-01093-f001:**
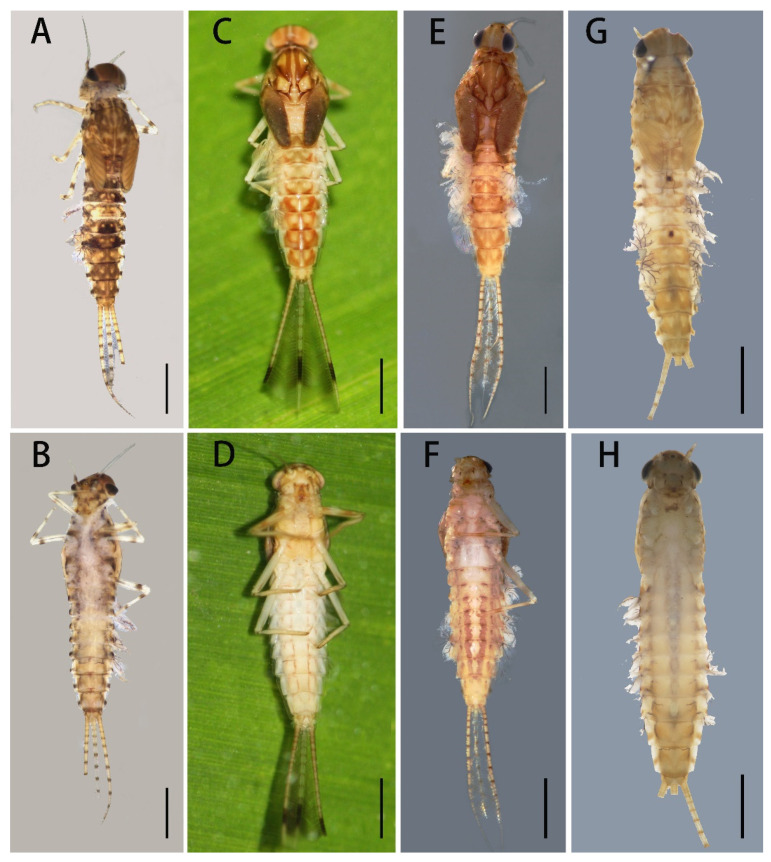
Nymphal habitus of four *Cloeon* species of China: (**A**) dorsal view of *C. bicolor*; (**B**) ventral view of *C. bicolor*; (**C**) dorsal view of *C. dipterum*; (**D**) ventral view of *C. dipterum*; (**E**) dorsal view of *C. harveyi*; (**F**) ventral view of *C. harveyi*; (**G**) dorsal view of *C. viridulum*; (**H**) ventral view of *C. viridulum*. Scale bars = 1.0 mm.

**Figure 2 insects-12-01093-f002:**
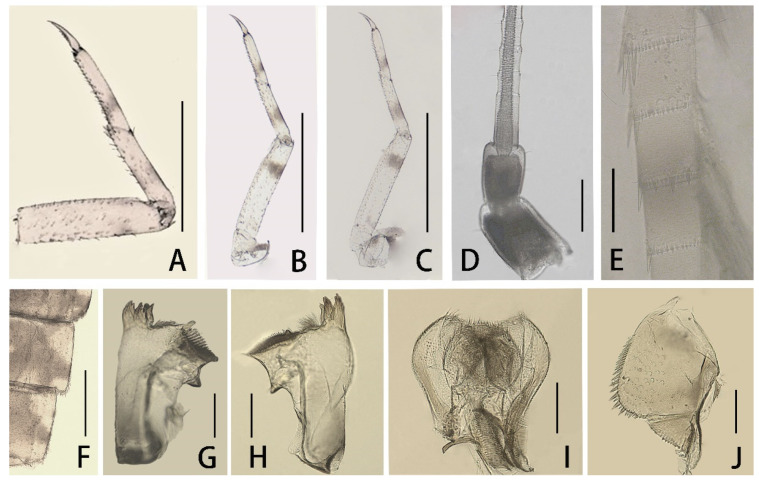
Nymphal structures of *C. bicolor*: (**A**) foreleg; (**B**) midleg; (**C**) hindleg; (**D**) antenna; (**E**) partial cerci; (**F**) lateral margins of terga VIII–IX; (**G**) left mandible; (**H**) right mandible; (**I**) hypopharynx; (**J**) paraproct. Scale bars: A–F = 0.5 mm; G–J = 0.1 mm.

**Figure 3 insects-12-01093-f003:**
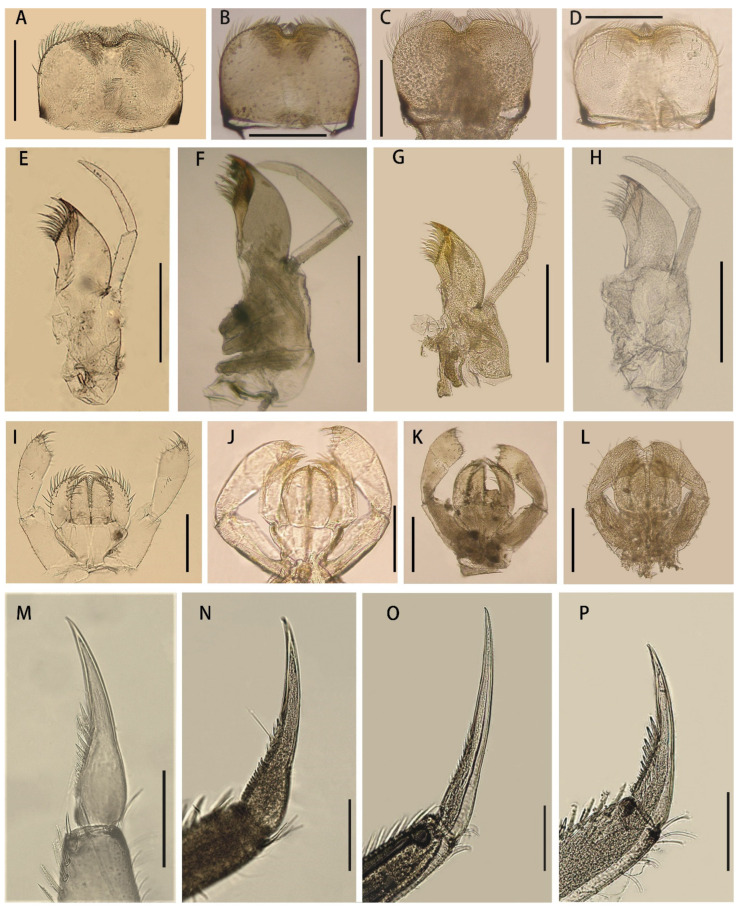
Mouthparts and claws of four Chinese *Cloeon* species: (**A**) labrum of *C. bicolor*; (**B**) labrum of *C. dipterum*; (**C**) labrum of *C. harveyi*; (**D**) labrum of *C. viridulum*; (**E**) maxilla of *C. bicolor*; (**F**) maxilla of *C. dipterum*; (**G**) maxilla of *C. harveyi*; (**H**) maxilla of *C. viridulum*; (**I**) labium of *C. bicolor*; (**J**) labium of *C. dipterum*; (**K**) labium of *C. harveyi*; (**L**) labium of *C. viridulum*; (**M**) claw (foreleg) of *C. bicolor*; (**N**) claw (foreleg) of *C. dipterum*; (**O**) claw (foreleg) of *C. harveyi*; (**P**) claw (foreleg) of *C. viridulum*. Scale bars = 0.1 mm.

**Figure 4 insects-12-01093-f004:**
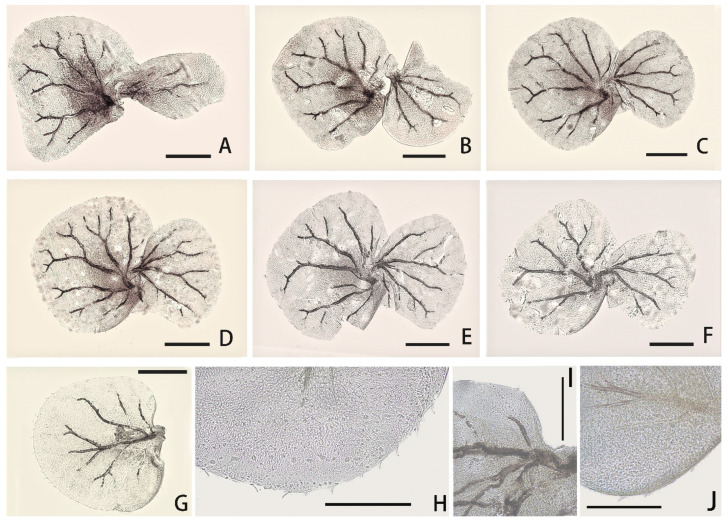
Gills of *C. bicolor*: (**A**) gill I; (**B**) gill II; (**C**) gill III; (**D**) gill IV; (**E**) gill V; (**F**) gill VI; (**G**) gill VII; (**H**) marginal setae of gill VII; (**I**) sclerotized inner margin of gill VII; (**J**) sclerotized outer margin of gill VII. Scale bars = 0.1 mm.

**Figure 5 insects-12-01093-f005:**
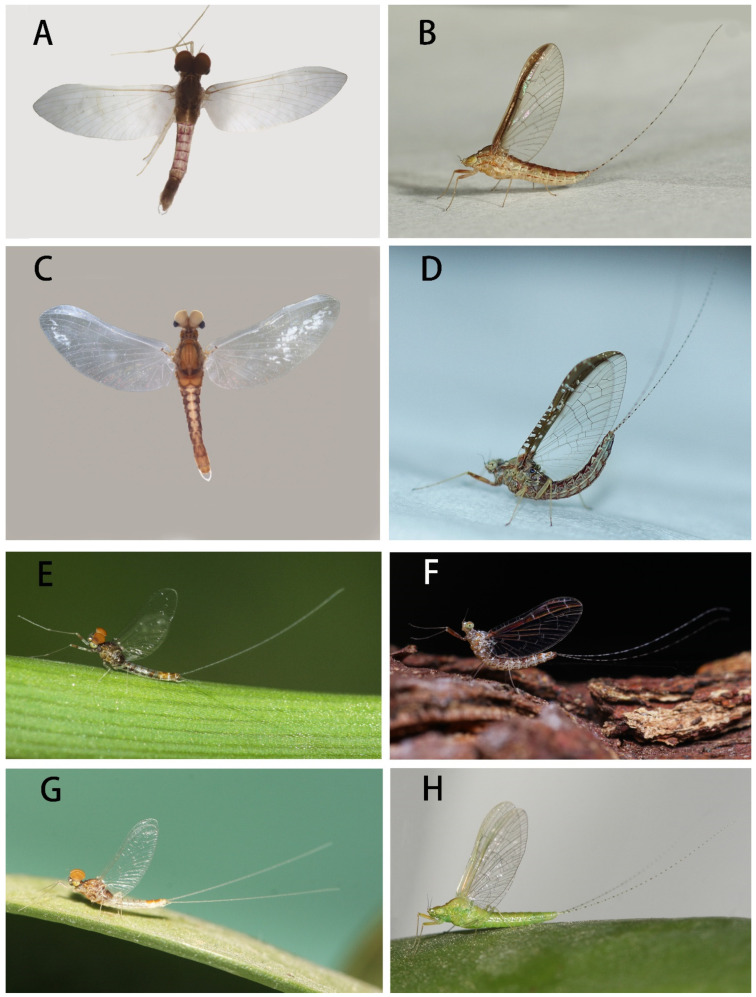
Adult habitus of four *Cloeon* species from China: (**A**) male of *C. bicolor*; (**B**) female of *C. bicolor*; (**C**) male of *C. dipterum*; (**D**) female of *C. dipterum*; (**E**) male of *C. harveyi*; (**F**) female of *C. harveyi*; (**G**) male of *C. viridulum*; (**H**) female of *C. viridulum*.

**Figure 6 insects-12-01093-f006:**
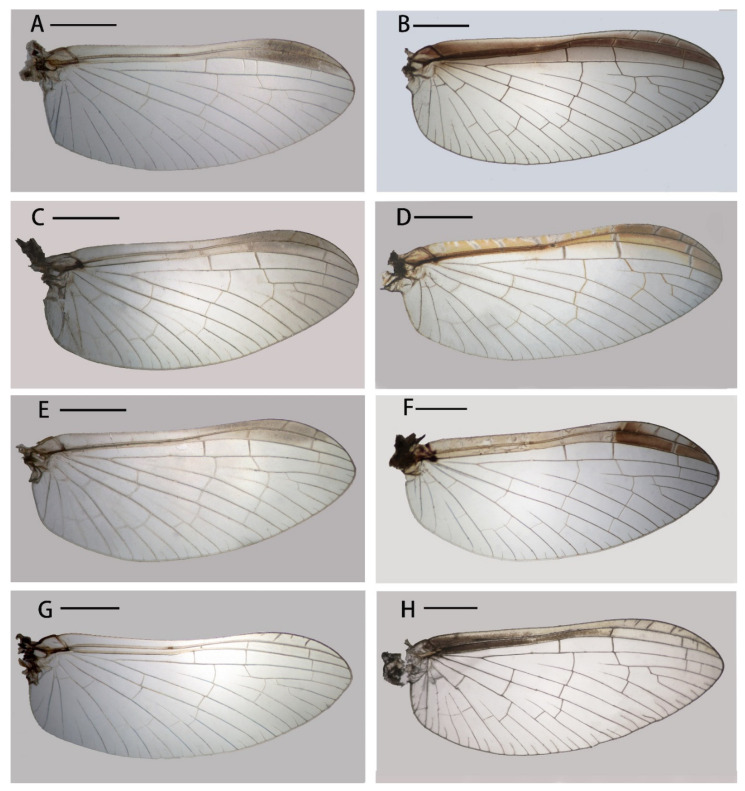
Wings of four *Cloeon* species of China: (**A**) male of *C. bicolor*; (**B**) female of *C. bicolor*; (**C**) male of *C. dipterum*; (**D**) female of *C. dipterum*; (**E**) male of *C. harveyi*; (**F**) female of *C. harveyi*; (**G**) male of *C. viridulum*; (**H**) female of *C. viridulum*. Scale bars = 1.0 mm.

**Figure 7 insects-12-01093-f007:**
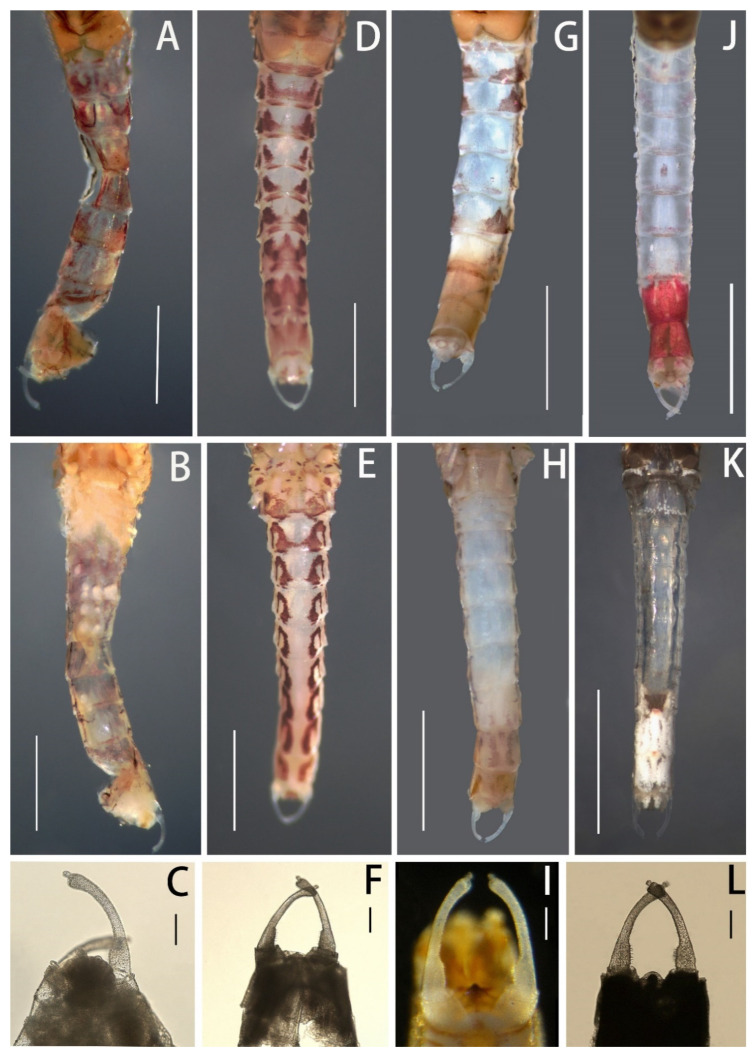
Male imaginal structures of four *Cloeon* species from China: (**A**) abdominal terga of *C. bicolor*; (**B**) abdominal sterna of *C. bicolor*; (**C**) genitalia of *C. bicolor* (ventral view); (**D**) abdominal terga of *C. dipterum*; (**E**) abdominal sterna of *C. dipterum*; (**F**) genitalia of *C. dipterum* (ventral view); (**G**) abdominal terga of *C. harveyi*; (**H**) abdominal sterna of *C. harveyi*; (**I**) genitalia of *C. harveyi* (ventral view); (**J**) abdominal terga of *C. viridulum*; (**K**) abdominal sterna of *C. viridulum*; (**L**) genitalia of *C. viridulum* (ventral view). Scale bars: A, B, D, E, G, H, J, K = 1.0 mm, C, F, I, L = 0.2 mm.

**Figure 8 insects-12-01093-f008:**
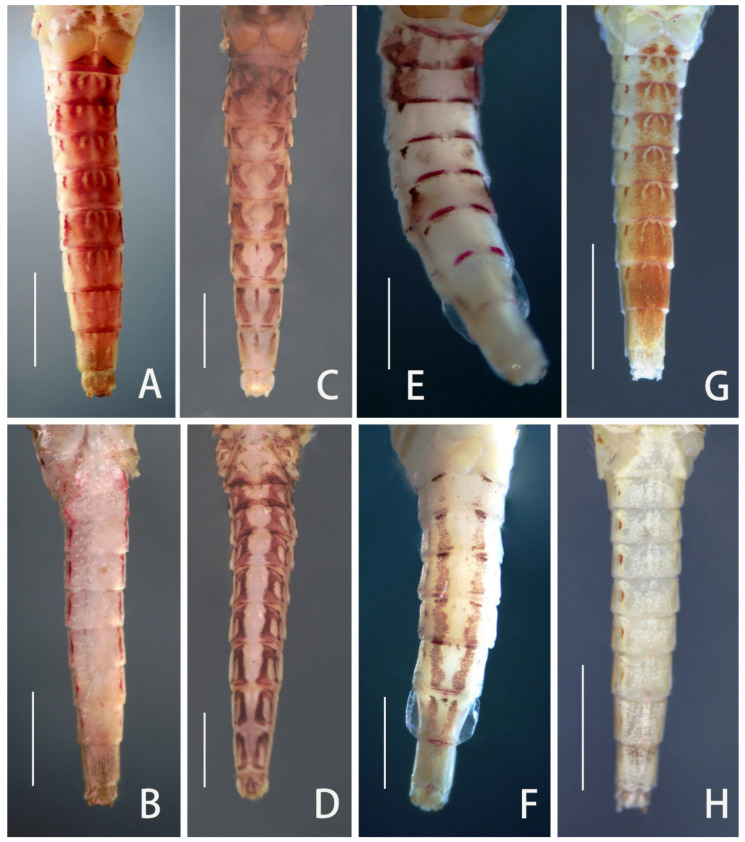
Female imaginal abdomens of four *Cloeon* species from China: (**A**) abdominal terga of *C. bicolor*; (**B**) abdominal sterna of *C. bicolor*; (**C**) abdominal terga of *C. dipterum*; (**D**) abdominal sterna of *C. dipterum*; (**E**) abdominal terga of *C. harveyi*; (**F**) abdominal sterna of *C. harveyi*; (**G**) abdominal terga of *C. viridulum*; (**H**) abdominal sterna of *C. viridulum*. Scale bars = 1.0 mm.

## Data Availability

All data is available in this paper.

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
