# Peer review of "A Review of the Genus Cloeon from Chinese Mainland (Ephemeroptera: Baetidae)"

_insects, 2021, doi:10.3390/insects12121093_

Round 1
Reviewer 1 Report
In this article, the authors provide the first comprehensive review of the genus Cloeon in China. This is a very important and useful work as part of the species are still poorly know (badly described and almost never reported since their original description). Moreover, they describe and illustrate the larval stage of Cloeon bicolor. Besides the numerous small details directly indicated in the ms, I have the main central concerns:
-The authors have to explain their concept of Cloeon. How do they differentiate Cloeon from related genera such as Procloeon at the different stages and sex? Do they consider Similicloeon as a valid genus or subgenus? Are Procloeon and Similicloeon also occur in China? What is the risk of confusion (high for larvae and male imago!)?
- The paper is mainly focused on China, but species from South Asia must be considered potentially present at least for the South of China. What are the potential species?
- Cloeon female imagos are rather easy to identify on pictures. Why the authors don’t use website such as iNaturalist to complete the distribution of the species and consider species potentially present?
- Part of the pictures are of good quality. Plates with comparison of the same structure between species are very important and generally well presented. But other pictures are clearly of insufficient quality. Essential characters such as the denticles on claw, shape of the subgenital plates on male forceps, shape of the labial palp are either not clearly visible or out of focus. Moreover the colour balance is not consistent between the different pictures of the same plate (see remarks in the text)
- The new synonymies need stronger arguments than just one line. I don’t say that they are not correct, but they are NOT ACCEPTABLE as presented. Detailed comparison must be made, not only stated that the two species have dark spots at the same place of the terga or both present broad distribution. Type material of Cloeon micki must be directly compared with specimens of Cloeon viridulum at ALL stages. Sometimes species can be separated only one stage. As long as strong arguments are not detailed, the synonymies remain doubtful.
- Cloeon dipterum is complex of closely related species (see the reconstructions of Rutschmann et al. 2017). How the authors can be sure that they have the true C. dipterum. Species from Japan close to C. dipterum, especially Cloeon ryogokuensis must be considered. Alternatively, the authors can solely indicate Cloeon cf dipterum.
- The key is very welcome. Unfortunately, the chosen characters are often insufficient to reliably separate the species, other characters are not used but are stronger and easier to use. The key must be rethought and rewritten
- Affinity section is a mix a speculation shared characters and ecological and behaviour data. I strongly suggest that just the ecological and behaviour data are kept and renamed the section.
- I wonder why barcoding of the four species are not provided? It is of first importance for large survey using molecular data.
- I am not English native, but at least some sentences sound odd and must be carefully checked
Because of the high interest of the topic, I highly encourage the publication of the manuscript after major revision as stated above
Jean-Luc Gattolliat

Reviewer 2 Report
Overall, this is a well-written and valuable contribution to the study of the mayfly genus Cloeon. I recommend it for publication after minor revisions.
Line 37: Rather than citing Edmunds et al. 1976 for North American Cloeon, I suggest the following:
Randolph, R. P., W. P. McCafferty, D. Zaranko, L. M. Jacobus and J. M. Webb. 2002. New Canadian records of Baetidae (Ephemeroptera) and adjustments to North American Cloeon. Entomological News 113: 306-308.
Also, line 37: Cloeon species have been found in South America, too. See the following citations:
Banegas, B. P., Túnez, J. I., Nieto, C., Fañani, A. B., Casset, M. A., & Rocha, L. (2020). First record of Cloeon dipterum (L.)(Ephemeroptera: Baetidae) in Buenos Aires, Argentina. Revista de la Sociedad Entomológica Argentina, 79(3), 24-33.
Salles, F. F., Gattolliat, J. L., Angeli, K. B., De-Souza, M. R., Gonçalves, I. C., Nessimian, J. L., & Sartori, M. (2014). Discovery of an alien species of mayfly in South America (Ephemeroptera). ZooKeys, (399), 1.
Lines 37-38: Cloeon also is known from various Pacific Islands, but much more work needs done on the identification and taxonomy. If you can add some ciations, that would be very helpful. Otherwise, see, for example:
Gattolliat, J. L., & Staniczek, A. (2011). New larvae of Baetidae (Insecta: Ephemeroptera) from Espiritu Santo, Vanuatu. Stuttgarter Beiträge zur Naturkunde A, Neue Serie, 4, 75-82.
Line 42: Use “sexes” instead of genders.
On page 2, line 51: “1930’” should have the apostrophe removed.
In the Materials and Methods (line 66), please give more details about the rearing techniques. Sometimes, different mayflies require different techniques for success.
Were slides temporary or permanent? What medium was used for mounting? Tell about this near Line 70.
Line 171: “fade and are not very typical” I am not sure what this means. Does this mean that some individuals have weak coloration? If so, be clear that his represents variation. You might consider a small section about morphological and color variation for each species. This is helpful for understanding your current species concepts and for recognizing species boundaries in the future.
Line 201: I would add “and complete synonymy” after “More citations”.
Lines 204, 245 (maybe elsewhere, too): I would say “stored” or “preserved” rather than deposited.
Line 222: Consider rephrasing to say “variation between populations” or “intraspecific variation”.
Lines 260-261: Did you observe any differences in the timing of emergence? Did you observe or note any other biological or habitat observations?
Line 319: Change “regarded as the juniour synonymy of” to “regarded as a junior synonym of”.
Line 385: Please be more descriptive about the key. I suggest “Keys to Cloeon species of mainland China”.
Lines 386, 397: Please add “adults”.
Reviewer 3 Report
Manuscript ID insects-1486635
The Manuscript reviews the genus Cloeon (Ephemeroptera : Baetidae) found in China and presents descriptions of four species and identifies new synonyms and a new combination based on extensive collections throughput China. Three species remain uncertain due to limited material.
Generally, the manuscript is of reasonable quality, but it would have improved greatly if the manuscript had been read by someone with English as their first language. The English expression at times is difficult to understand and needs extensive revision. I have made some suggestions below.
Specific Comments:
Simple Summary:
L10 insert ‘the’ before world.
L12–16 In this study of over one thousand specimens from over 50 sites on the Chinese mainland for species of Cloeon were identified, namely, C. bicolor Kimmins ,1947, C. dipterum (Linnaeus, 1761), C. harveyi (Kimmins, 1947) and C. viridulum Navas 1931. The record of C. bicolor is the first for China and its nymphs are described for the first time. Three junior synonyms are proposed for the species C. viridulum. [I can only find 3 synonyms for this species]
L17 ‘solid base for future works’
Abstract
L20–21 After examining the types of two species (which ones were they?). If they were only C. viridulum and C. apicatum these are only sufficient to justify synonymy of the two under C. viridulum. It would have been useful to examine all the types that were available or still exist.
L24 Delete ‘confirmed upon specimens’ and replace with ‘also recognised’
L26 Delete ‘to the fourth species’ and replace with ‘with’
L27–28 You cannot consider C. virens Klapalek, 1905 as a junior synonym of C. viridulum when it appears that you propose it as a misidentification by Ulmer of C. viridulum from China. If you think C.virens and C. viridulum are synonyms then C. virens has priority over C. viridulum under the Code because it was described some 26 years earlier that is, C. viridulum would be the junior synonym.
- Introduction
L35 Delete ‘The’ and start sentence with Cloeon
L38 Suggest – ‘However, the species in this genus have never been systematically studied from China.’
L41–54 ‘They describe subimagoes, females or even undeclared genders but the types are lost or damaged ([16,17] and the descriptions are usually extremely brief and often lack figures. Consequently, these poorly described species remain an impediment to a clear understanding of the species found in China. The ability to recognise new species (eg. Cloeon micki Tong and Dudgeon, 2021 [4] and those previously described is not clear.’
L48 Suggest – ‘In order to solve this problem, we collected and reared Cloeon species from more than 50 water bodies on the Chinese mainland. We examined photos of type material (C. viridulum and C. apicatum from Huede Museum deposited by Navas and Hsu in 1930. From this material we were able to recognise four species from the Chinese mainland which were photographed and descriptions were updated.’ [Delete Line 51–53 as this should go in acknowledgements, not in the main text. Figure 9 is also not required or particularly useful.
L55–58 Unfortunately, three species from the Taiwan Province (list of species as in L55–56) were limited in our collections and require further material to determine their status in China. Until this is done they must remain as Incertaae sedis.
Materials and Methods
Too brief and does not include any specific details, such as type of habitats sampled, mesh size of nets, type pf light traps used, clear light, UV or what, type of stereo microscope used, mounting medium used for slide preparation, type of microscope camera used etc.
Results
L99 Delete ‘one’
L100 Delete ‘one’
L117 Delete ‘this makes different terga with different color pattern’
L119 Delete ‘all’ and ‘too’
L121 Delete ‘this’
L135 Change ‘castaneous’ to chestnut
L137 Change ‘this makes terga look like with’ to ‘producing’ or ‘giving’
L138 Change ‘midlines’ to ‘midline’
L141–42 Need description of forceps
L155–166 This is all discussion or remarks
L155 Delete ‘counterparts of’
L156 Change ‘only’ to ‘having’
L157 Delete ‘tails’ and replace with ‘caudal filaments’
L158 In this line you say ‘they can be identified’ but to which species does ‘they’ refer, C. diperum or C. bicolor? If C. dipterum then say so. Eg ‘C. dipterum has pale abdominal sterna…’ and delete ‘However, they can be identified by’
L165 Delete ‘meanwhile without’ and replace with ‘but lacking’
L173 Replace ‘dots’ with ‘spots’. Also change throughout manuscript for C. viridulum
L176 Material examined. For an international journal not to include latitude and longitude for each location is not acceptable, particularly for readers from all other countries.
L204 Delete ‘deposited’, as it is redundant. Could replace with ‘preserved’, but not necessary. Fix this for all species that follow.
L208 Delete ‘of it without’ and replace with ‘lacking’
L209 Insert ‘at’ after ‘except’
L210 Delete ‘of it’
L213 Replace ‘diagnosed’ by ‘recognised’
L215 Insert ‘of’ between ‘1/2 and claw’
L219 insert ‘the’ between ‘on and lateral’ and insert ‘although’ between ‘only and some’
L220 Delete ‘their’ and replace with ‘the’. Also, insert ‘the’ between ‘on and basal’
L220 Delete ‘the’
Latitudes and longitudes required for locations.
L250 Insert ‘the’ before ‘subcostal brace’
L251 Delete ‘of it’
L252 Delete ‘those’ and replace with ‘the’
L254 Delete ‘stigma and replace with ‘pterostigma’
L258 Delete ‘close’ and replace with ‘similar’
L260 Insert ‘the after ‘sharing’
Latitudes and longitudes required
L300 As for L204
L304 Insert a comma after ‘dorsally’ and delete ‘and’
L312–313 Delete whole sentence starting with ‘Sometimes’
L314 Delete ‘dots’ and replace with ‘spots’
The description by Tong and Dudgeon is excellent and could be considered as a template for the description in this manuscript.
L314–315 Although the types of C. micki Tong and Dudgeon have not been examined the presence of two spots on abdominal terga II and V in all life stages (occasionally not obvious) the similar male genitalia, dorsal color pattern of males and females and characteristics of the tarsal claws and mouthparts of the nymphs support the designation of C. micki as a junior synonym of C. viridulum.
L315–317 As mentioned above you cannot designate C. virens as a junior synonym as it was described first. It appears to be a misidentification by Ulmer, so not a junior synonym at all.
L319 Junior is misspelt in text.
L319 Delete ‘Besides’ and start sentence ‘The types of both species …’
L321–322 Since the type of C. pielinum is lost and the description is poor you need evidence for a synonymy. You do not present any so it is not possible to create a synonymy but rather this species should be designated as Incertae sedis.
Latitudes and longitudes
L362 Insert ‘based on’ before ‘abdominal’
L366 Change ‘same’ to similar’
L372–373 Delete last sentence and replace with ‘C. harveyi and C. dipterum are also slightly darker than C. viridulum and C. bicolor.’
L374–L384 Delete
L395 Delete ‘Except terga VII–IX, terga pale’ and replace with ‘Terga I–VI pale, terga VII–IX pigmented;
L402 Replace ‘stigma’ with ‘pterostigma’
L403 Figure 6F not 8F
L409 Insert ‘anterior’ after median
Acknowledgements
Include acknowledgements for figure 9.
L432 Replace ‘help’ with ‘helped’
Figures
Quality of figures is variable with some ie. Figure 1, 5, 6, 7, 8 excellent but the variation of backgrounds and contrast for Figures 2, 3, 4 and 7 needs to be improved to clearly see features.
Pity that Figure 7J and Figure 8G do not show abdominal spots of C. viridulum, especially as it is diagnostic and used to synonymise C. micki.
There is no need for Figure 9 other than to confirm you saw photos of types. Delete figure.
L451 Journal title to be spelled out in full and in italics.
L520 Why is ‘Huang, B.K. (ed.) in italics?
Round 2
Reviewer 1 Report
I agree with most of the answers of the authors and the modifications are generally accurate.
Two points still need to be fixed.
- The concept of Cloeon used by the authors need to be clearly stated in the introduction. Especially what they consider as belonging to Cloeon and what does? based on which morphological characters?
- The authors don't want to include molecular evidence in the article (which I consider as a mistake), but they have at least to discuss some important cases: the sequence KR612252 deposited on NCBI by Zhou D. and Zhou C. identified as Clooen viridulum match at 99% Cloeon ryogokuensis from Japan. Does it mean that Cloeon ryogokuensis is present in China or alternatively that Cloeon ryogokuensis is synonym of Cloeon viridulum?
- I don't believe that the size of the specimens is a justification for not sequencing them. Even a leg of Cloeon is sufficient for DNA extraction and amplification. If the authors don't have to add molecular data, they have to indicate it clear and explain what the potential of using integrative approach in such a context
